# Over-time changes in the prevalence of metabolic syndrome and its components among elderly population in Iran from 2016 to 2021; A nation-wide study

Seyed Mohammad-Navid Ataei[1☯‡], Ali Sheidaei[1☯‡], Ali Golestani[1], Sepehr Khosravi[1], Mohammad-Mahdi Rashidi[1], Ozra Tabatabaei-Malazy[1,2*], Rosa Haghshenas[1], Kazem Khalagi[3,4], Bagher Larijani[2]

1 Non-Communicable Diseases Research Center, Endocrinology and Metabolism Population Sciences Institute, Tehran University of Medical Sciences, Tehran, Iran, 2 Endocrinology and Metabolism Research Center, Endocrinology and Metabolism Clinical Sciences Institute, Tehran University of Medical Sciences, Tehran, Iran, 3 Obesity and Eating Habits Research Center, Endocrinology and Metabolism Clinical Sciences Institute, Tehran University of Medical Sciences, Tehran, Iran, 4 Osteoporosis Research Center, Endocrinology and Metabolism Clinical Sciences Institute, Tehran University of Medical Sciences, Tehran, Iran

☯ These authors contributed equally to this work.
‡ These authors share first authorship on this work.
* tabatabaeiml@sina.tums.ac.ir

## Abstract

### Background

The prevalence of Metabolic Syndrome (MetS) increases with aging, significantly contributing to the rising burden of non-communicable diseases (NCDs). This study aimed to investigate over-time changes in the prevalence of MetS and its components among the elderly population of Iran.

### Methods

We analyzed data from the 2016 and 2021 national STEPwise approach to non-communicable disease risk factor Surveillance (STEPS) for participants aged ≥65 who completed all three survey steps (questionnaire-based assessments, physical measurements, and laboratory tests) with no missing data on MetS components. According to the STEPS protocol, individuals with severe mental disorders, physical limitations that prevented measurements, or inability to provide laboratory samples were excluded. Using survey analysis techniques and weights, we ensured the findings are nationally representative. MetS was defined using the following criteria: National Cholesterol Education Program Adult Treatment Panel III (NCEP ATPIII), International Diabetes Federation (IDF), American Heart Association/National Heart, Lung, and Blood Institute (AHA/NHLBI), and Joint Interim Statement (JIS). Regional

**Data availability statement:** The presented tables/figures are without any personal information. As we said above, the datasets generated and/or analyzed during the current study are not publicly available due to the restrictions set by the funder of the main STEPS project, I.R. Iran's National Institute of Health Research. Interested researchers may request data access via the following official point of contact: Email: nihr@tums.ac.ir, Website: https://nihr.tums.ac.ir. This contact ensures long-term and institutional oversight of data access. Data are securely stored and maintained under institutional data management policies.

**Funding:** OTM received the grant. This work was supported by the Non-Communicable diseases Research Center, Endocrinology and Metabolism Population Sciences Institute, Tehran University of Medical Sciences, Tehran, Iran [Grant IDs 1401-4-221-63616]. The funders had no role in study design, data collection and analysis, decision to publish, or preparation of the manuscript.

**Competing interests:** The authors have declared that no competing interests exist.

**Abbreviations:** AHA/NHLBI: American Heart Association/National Heart, Lung, and Blood Institute; APC: Annual percent change; BMI: Body mass index; CI: Confidence interval; DALYs: Disability-adjusted life years; DBP: Diastolic blood pressure; FPG: Fasting plasma glucose; HC: Hip circumference; HDL-C: High-density lipoprotein cholesterol; IDF: International Diabetes Federation; JIS: Joint Interim Statement; LDL-C: Low-density lipoprotein cholesterol; MetS: Metabolic Syndrome; NCDRC: Non-Communicable Diseases Research Center; NCDs: Non-communicable diseases; NCEP ATPIII: National Cholesterol Education Program Adult Treatment Panel III; NHANES: National Health and Nutrition Examination Survey; NIHR: National Health Research Institute of Islamic Republic of Iran; RIDF: Regional IDF; RJIS: Regional JIS; SBP: Systolic blood pressure; STEPS: STEPwise approach to non-communicable disease risk factor Surveillance; TG: Triglycerides; WC: Waist circumference.

IDF (RIDF) and regional JIS (RJIS) were defined by ethnicity-specific values of waist circumference.

## Results

This study included 4,000 elderly participants in 2016 and 3,849 in 2021, with a mean age of 74 and 72 years, respectively. Approximately 50% were female in both years, and the proportion of urban residents increased from 67% in 2016 to 75% in 2021. The national prevalence of MetS among the elderly increased significantly from 46.11%−66.38% in 2016 to 54.15%−73.98% in 2021 across different criteria, with an annual percent change of 2.19%−3.26%. Prevalence was higher in females and urban populations, while males showed a greater increase. High blood pressure, low high-density lipoprotein cholesterol, and central obesity were the most common components of MetS, while elevated triglycerides and high fasting plasma glucose showed the largest increases over time.

## Conclusions

The escalating prevalence of MetS in the elderly is a matter of increasing concern, underscoring the need for targeted policy interventions aimed at promoting healthy aging, empowering the elderly, and advocating for lifestyle modifications.

## Introduction

Non-communicable diseases (NCDs) have become the leading cause of morbidity and mortality worldwide. This trend is driven by rapid urbanization, population aging, and globalization of unhealthy lifestyles, such as poor dietary habits and sedentary behavior [1]. Cardiovascular diseases (particularly ischemic heart disease and stroke) along with diabetes, are among the top tencontributors to the global burden of NCDs, disproportionately affecting older adults [2]. Metabolic syndrome (MetS), also referred to as syndrome X or the insulin resistance syndrome, is a cluster of metabolic disorders thatfrequently co-occur, increasing the risk of type 2 diabetes and cardiovascular diseases. These disorders include high blood pressure (HBP), impaired fasting plasma glucose (FPG), central obesity, hypertriglyceridemia, and low high-density lipoprotein cholesterol (HDL-C) [3].

Aging is a significant risk factor for the development and progression of metabolic disorders in older adults, primarily due to various physiological changes such as decreased metabolic rate, hormonal alterations, and shifts in body composition. Additionally, the higher prevalence of physical inactivity and cumulative lifetime exposure to environmental factors further contribute to this risk [2,3]. Between 2000 and 2021, global health challenges shifted. While there was a decline in disability-adjusted life years (DALYs) attributable to behavioral and environmental risks, DALYs due to metabolic risks increased by 49.4%, having a greater impact on older adults. The burden increased considerably due to both rising risk exposure and an aging population [4].

Mortality rates related to MetS are disproportionately higher in low- and lower-middle-income countries [5]. Type 2 diabetes, a major outcome of MetS, also imposes the greatest burden in North Africa and the Middle East [6]. The global prevalence of MetS ranges from 12.5% to 31.4%, depending on the criteria used for its definition, with a higher prevalence in the Eastern Mediterranean Region and Americas [7]. This prevalence varies across different countries, ethnicities, genders, and age groups. However, studies indicate that MetS shows an increasing trend in both developed and developing countries [8–10].

Iran, a lower-middle-income country in the Eastern Mediterranean Region, has implemented a national survey program following the World Health Organization's STEPwise approach to NCD risk factor surveillance (STEPS), which has been conducted in eight rounds since 2005 [11]. According to the 2016 national survey, the prevalence of MetS in Iran was alarmingly high, ranging from 32% to 47.6% based on various definitions. The prevalence was particularly higher among females, urban residents, and the elderly [12]. However, there is limited published evidence on the more recent prevalence of MetS in Iran and no nationally representative study focusing on the elderly population has examined the changes in recent years.

Studying the prevalence and trend of MetS as a potential indicator for identifying individuals at higher risk of NCDs is crucial, especially in the elderly population, which bears the highest burden. To date, few studies have examined temporal changes in MetS prevalence among the elderly in Iran using nationally representative data. Moreover, subnational differences and trends based on standardized definitions of MetS remain underexplored in this high-risk group. In this study, we aim to investigate the national and sub-national prevalence of MetS and its components, based on different definitions, in the elderly population of Iran. We used data from STEPS 2021 and compared it with STEPS 2016 to assess changes over time. The findings of this study could be useful for countries in the region or those with similar income levels.

## Methods and materials

### Study design

We used data from the STEPS national surveys conducted in 2016 and 2021. STEPS is a large-scale national survey program, conducted in Iran according to the STEPwise approach proposed by the WHO, aiming to monitor NCD risk factors [13]. All Iranian adults aged ≥18 years from urban and rural areas of all provinces of Iran were included in these surveys using a stratified random cluster sampling method. The STEPS surveys comprised three stages: (1) questionnaire-based assessments to investigate sociodemographic status and medical history; (2) physical measurements to assess anthropometric variables, blood pressure, and pulse rate; and (3) laboratory measurements. Individuals were excluded from the study if they had mental disorders that impaired their ability to respond to questionnaires, had physical limitations preventing anthropometric measurements, were unable to provide laboratory samples, or were pregnant. More details on the methodology of these surveys can be found elsewhere [11,14].

### Data collection and measurements

To achieve a nationally and subnationally representative sample size, a systematic cluster sampling method was used. Clusters were determined based on the population size of each province and relative weighs. This resulted in the selection of 3105 clusters for STEPS 2016 and 3167 clusters for STEPS 2021, each comprising 10 participants. Data collection for STEPS 2016 started on June 21, 2016, and ended on September 22, 2016. For STEPS 2021, data collection began on January 10, 2020, but was suspended on February 1, 2020, due to the onset of the COVID-19 pandemic. The first 318 clusters of STEPS 2021 were completed before the COVID-19 pandemic. However, due to pandemic-related modifications, the number of participants in each of the 2849 remaining clusters was reduced to 9.Data collection operations for STEPS 2021 were resumed on January 10, 2021, and completed by April 30, 2021. A limited number of pilot questionnaires were administered in the autumn of 2020.

All anthropometric variables and blood pressure were measured according to WHO recommended protocols [15], using standardized and calibrated instruments of the same brands. Height was measured while standing barefoot, with heels, buttocks, and back of the head aligned against a wall. Weight was measured using a digital scale. Body mass index (BMI) was calculated as weight (kg) divided by height in squared meters ($m^2$). Waist circumference (WC) and hip circumference (HC) were measured while the subjects were not wearing restrictive underwear and heavy pants or skirts. Systolic and diastolic blood pressures (SBP and DBP) were measured on the brachial artery by standard Beurer sphygmomanometers after a 15-minute-rest in sitting position, three times, three minutes apart. The mean of the last two measurements was considered as the final value. Serum levels of FPG, HDL-C, and triglycerides (TG) were measured using an auto-analyzer (Roche-Hitachi Cobas C311, High–Technologies Corporation, Tokyo, Japan) at the central laboratory in Non-Communicable Diseases Research Center (NCDRC), Tehran University of Medical Sciences, Tehran, Iran.

## Diagnostic criteria for MetS

We defined MetS according to four different criteria, including the National Cholesterol Education Program Adult Treatment Panel III (NCEP ATPIII) [16], the International Diabetes Federation (IDF) [17], the American Heart Association/National Heart, Lung, and Blood Institute (AHA/NHLBI) [18], and the Joint Interim Statement (JIS) [19]. The Iranian National Committee of Obesity recommended a uniform WC cut-off of 95 cm for both men and women. This cut-off was applied to the IDF and JIS definitions, resulting in two modified criteria, known as Regional IDF (RIDF) and Regional JIS (RJIS) [20]. The details on all six criteria are represented in Table 1.

## Statistical analysis

In this study, we included subjects aged 65 years or older who participated in all three phases of STEPS and had no missing data on the variables required to define MetS. We calculated national and subnational prevalence of MetS according to different criteria for the entire elderly population and different subgroups based on sex and place of residence (rural or urban areas)

**Table 1. Diagnostic criteria used to define MetS.**

| Components | NCEP ATPIII | IDF | RIDF | AHA/NHLBI | JIS | RJIS |
|---|---|---|---|---|---|---|
| **Glucose domain** | FBS ≥ 100 mg/dl (Includes diabetes) | FBS ≥ 100 mg/dl (Includes diabetes) | FBS ≥ 100 mg/dl (Includes diabetes) | FBS ≥ 100 mg/dl (Includes diabetes) | FBS ≥ 100 mg/dl (Includes diabetes) | FBS ≥ 100 mg/dl (Includes diabetes) |
| **Obesity domain** | WC ≥ 102 cm (men)/ ≥ 88 cm (women) | WC ≥ 94 cm (men)/ ≥ 80 cm (women) | WC ≥ 95 cm (for Iranian men & women) | WC ≥ 102 cm (men)/ ≥ 88 cm (women) | WC ≥ 94 cm (men)/ ≥ 80 cm (women) | WC ≥ 95 cm (for Iranian men & women) |
| **Lipid profile** | TGs ≥ 150 mg/dl | TGs ≥ 150 mg/dl or under treatment | TGs ≥ 150 mg/dl or under treatment | TGs ≥ 150 mg/dl or under treatment | TGs ≥ 150 mg/dl or under treatment | TGs ≥ 150 mg/dl or under treatment |
| | HDL-C < 40 mg/dl (men)/ < 50 mg/dl (women) | HDL-C < 40 mg/dl (men)/ < 50 mg/dl (women) or under treatment | HDL-C < 40 mg/dl (men)/ < 50 mg/dl (women) or under treatment | HDL-C < 40 mg/dl (men)/ < 50 mg/dl (women) or under treatment | HDL-C < 40 mg/dl (men)/ < 50 mg/dl (women) or under treatment | HDL-C < 40 mg/dl (men)/ < 50 mg/dl (women) or under treatment |
| **Blood Pressure domain** | SBP ≥ 130 mmHg or DBP ≥ 85 mmHg | SBP ≥ 130 mmHg or DBP ≥ 85 mmHg or Under treatment | SBP ≥ 130 mmHg or DBP ≥ 85 mmHg or Under treatment | SBP ≥ 130 mmHg or DBP ≥ 85 mmHg or Under treatment | SBP ≥ 130 mmHg or DBP ≥ 85 mmHg or Under treatment | SBP ≥ 130 mmHg or DBP ≥ 85 mmHg or Under treatment |
| **MetS definition** | Any 3 of the above components | Obesity domain plus any 2 of the above components | Obesity domain plus any 2 of the above components | Any 3 of the above components | Any 3 of the above components | Any 3 of the above components |

NCEP ATPIII National Cholesterol Education Program Adult Treatment Panel III; **IDF:** International Diabetes Federation; **RIDF:** Regional IDF; **AHA/NHLBI:** American Heart Association/National Heart, Lung, and Blood Institute; **JIS:** Joint Interim Statement; **RJIS:** regional JIS; **FBS:** Fasting Blood Sugar; **WC:** Waist circumference; **TGs:** Triglycerides; **HDL-C:** High Density Lipoprotein Cholesterol; **SBP:** Systolic Blood Pressure; **DBP:** Diastolic Blood Pressure; **MetS:** Metabolic Syndrome.

in 2016 and 2021. Furthermore, we estimated the prevalence of each component (central obesity, elevated TG, reduced HDL-C, elevated FPG and BP) in the population identified for MetS based on different criteria. We reported the prevalence with 95% confidence intervals (CI), after weighting the samples for general non-response, step-specific non-response, and demographic factors (age, sex, and area of residence) based on the most recent Iran census in 2016. Eventually, we evaluated the annual percent change (APC) of prevalence during 2016–2021 with their 95% CI using the following formula:

$$\text{APC} = [(P_n/P_0)^{1/N} - 1] \times 100$$

where $P_n$ represents the prevalence at the later time period, $P_0$ is the prevalence at the earlier time period, and $N$ is the number of years in the interval. APCs were used to compare over-time changes in the prevalence of MetS and its components.

The data analyses were executed using R statistical software version 4.3.2. A p-value of less than 0.05 was considered statistically significant.

### Ethical considerations

All participants voluntarily enrolled in each STEPS survey after being informed about the study's objectives and procedures, and providing written consent. Data were collected anonymously and computer-based at all stages of STEPS. Ethical approval for each STEPS survey was granted by the Ethical Committee of the National Health Research Institute of Islamic Republic of Iran (NIHR). Access to the dataset for the present study was granted on April 15, 2023, following ethics approval from the Ethical Committee of Endocrinology and Metabolism Research Institute, Tehran University of Medical Sciences, Tehran, Iran (IR.TUMS.EMRI.REC.1401.082).

## Results

### Participant characteristics

This study included 4000 participants aged 65 years or older in 2016 and 3849 participants in 2021. The mean age of participants was significantly lower in 2021 compared to 2016 (p<0.001). There were no statistically significant differences between the 2016 and 2021 data in terms of sex, systolic blood pressure, and low-density lipoprotein cholesterol (LDL-C) levels (p>0.05). However, there were statistically significant differences in the distribution of residential status, years of education, and other anthropometric and biochemical measures (p<0.001), as shown in Table 2.

### Overall prevalence of MetS

The overall estimated prevalence of MetS showed an increasing trend from 2016 to 2021 across various definitions. In 2016, the prevalence was 53.33% (95% CI: 51.03% − 55.61%) according to NCEP ATPIII, 59.80% (57.56% − 62%) according to IDF, 66.38% (64.24% − 68.45%) according to JIS, and 59.68% (57.43% − 61.89%) according to AHA/NHLBI. By 2021, the prevalence had increased to 60.12% (57.37% − 62.81%) for NCEP ATPIII, 67.64% (65.11% − 70.06%) for IDF, 73.98% (71.63% − 76.20%) for JIS, and 67.76% (65.13% − 70.28%) for AHA/NHLBI. The annual percent change (APC) in the overall prevalence of MetS in the elderly ranged from 2.19% to 2.57%, with the highest increase estimated using the AHA/NHLBI criteria (2.57%, 95% CI: 1.51% − 3.67%). Additionally, using regional definitions (RJIS and RIDF), the overall increasing trend in MetS prevalence remained consistent. Notably, the APC in overall prevalence was higher based on regional definitions: 3.26% (1.83% − 4.76%) for RIDF and 2.83% (1.80% − 3.91%) for RJIS (Table 3).

### Sex and residence differences

MetS prevalence was significantly higher among females than males and in urban compared to rural populations, across all definitions. The increase in MetS prevalence was not statistically significant in females, except when using the IDF

**Table 2.** Baseline characteristics of study participants.

| Variable | Category/ Unit | 2016 | 2021 | p-value |
|---|---|---|---|---|
| Age (years) | Mean (SD) | 74 (7) | 72 (6) | <0.001 |
| Sex | Female | 1,996 (50%) | 1,934 (50%) | 0.8 |
| | Male | 2,004 (50%) | 1,915 (50%) | |
| Residence | Rural | 1,314 (33%) | 960 (25%) | <0.001 |
| | Urban | 2,686 (67%) | 2,889 (75%) | |
| Years of education | Illiterate | 481 (12%) | 557 (15%) | <0.001 |
| | 1–6 years | 1,047 (26%) | 1,180 (31%) | |
| | 7–12 years | 461 (12%) | 249 (6.5%) | |
| | ≥12 years | 2,011 (50%) | 1,833 (48%) | |
| Self-reported medication for: | Diabetes | 579 (14%) | 713 (19%) | <0.001 |
| | Hypertension | 1,416 (35%) | 1,884 (49%) | <0.001 |
| | Dyslipidemia | 626 (16%) | 934 (24%) | <0.001 |
| Low Physical Activity | <600 METs | 2,507 (67%) | 2,245 (58%) | <0.001 |
| Anthropometric and lab measures | WC (cm) | 95 (14) | 97 (13) | <0.001 |
| | WHR | 0.95 (0.10) | 0.96 (0.11) | <0.001 |
| | Weight (kg) | 68 (14) | 69 (14) | <0.001 |
| | BMI (kg/m²) | 26.6 (5.0) | 27.1 (5.0) | <0.001 |
| | SBP (mmHg) | 141 (22) | 140 (21) | 0.3 |
| | DBP (mmHg) | 79 (12) | 80 (12) | <0.001 |
| | FBS (mg/dL) | 109 (42) | 113 (40) | <0.001 |
| | Total Cholesterol (mg/dL) | 167 (38) | 172 (41) | <0.001 |
| | Triglycerides (mg/dL) | 125 (69) | 145 (70) | <0.001 |
| | LDL-C (mg/dL) | 101 (33) | 100 (36) | 0.7 |
| | HDL-C (mg/dL) | 42 (12) | 43 (11) | <0.001 |

**METs** Metabolic Equivalent of Task score; **WC:** Waist Circumference; **WHR:** Waist to Hip Ratio; **BMI:** Body Mass Index; **SBP:** Systolic Blood Pressure; **DBP:** Diastolic Blood Pressure; **FBS:** Fasting Blood Sugar; **LDL-C:** Low Density Lipoprotein Cholesterol; **HDL-C:** High Density Lipoprotein Cholesterol

criteria (APC: 1.19%, 95% CI: 0.12% – 2.25%). However, males showed a significantly higher increase in prevalence than females according to all definitions, with the highest APC estimated using the AHA/NHLBI criteria (5.01%, 2.94% – 7.19%). There was no significant difference in the APC of MetS prevalence between rural and urban areas according to any of the definitions. According to regional definitions of RIDF and RJIS, females significantly showed increased prevalence, with an APC of 2.64% (95% CI: 0.65% – 4.63%) and 2.19% (0.87% – 3.51%), respectively. According to these regional definitions, differences between males and females were not statistically significant (Table 3).

## Sub-national disparities

The sub-national analysis of MetS prevalence in the elderly population of Iran reveals significant geographic disparities. The highest sub-national prevalence of MetS in the elderly, estimated by JIS, was 76.20% in 2016 and 83.90% in 2021, while the lowest, estimated by RIDF, was 23.31% in 2016 and 28.70% in 2021. Certain provinces, such as Bushehr and Khuzestan in 2016, and Ardebil and Gilan in 2021, showed particularly high prevalence rates. Conversely, regions like North-Khorasan and South-Khorasan consistently showed lower prevalence rates (Fig 1). APCestimates revealed significant increases in some provinces, especially among males in North-Khorasan and females in Sistan-and-Baluchistan (Fig 2). In line with national analysis, MetS prevalence in most provinces was higher in females. However, the significant rise in

MetS prevalence was more commonly observed in males than in females. Detailed prevalence and APC data by province, sex, and criteria can be found in S1 Table.

**Components of MetS**

As shown in Table 4, central obesity (2016: 75.47% – 100%; 2021: 75.57% – 100%), HBP (2016: 85.84% – 90.68%; 2021: 84.59% – 91.3%), and reduced HDL-C (2016: 80.48% – 89.8%; 2021: 76.07% – 88.45%) were the most frequent components of MetS in both years according to all definitions. Central obesity was present in all participants with MetS according to IDF and RIDF because these definitions mandate central obesity as a criterion. Elevated TG and elevated FPG showed the highest increases across all definitions, with the highest APC being 7.30% (95% CI: 4.97% – 9.79%) for TG according to IDF and 2.64% (1.45% – 3.88%) for FPG according to NCEP ATPIII. In contrast, HBP, central obesity, and reduced HDL-C remained stable. FPG increased more in urban areas and in females, while TG increased more in rural areas and in males across all definitions. However, these differences were not statistically significant.

## Discussion

This nationally representative study outlined the prevalence and temporal trends of MetS among Iranian adults aged ≥65 years from 2016 to 2021. The overall prevalence was considerably high across all diagnostic criteria, particularly among women and urban residents, with the highest estimate based on JIS criteria. A significant upward trend in MetS prevalence was observed over the study period, with a greater increase in men. The most prevalent components were HBP, low HDL-C, and central obesity. While these components remained consistently common, the largest increases over time were seen in elevated TG and high FPG.

Previous studies have shown that the prevalence of MetS increases with age. A meta-analysis of studies in Iran (2000–2016) showed a rise from 12.1% in the 20–29-year-olds to 51.7% in those over 60 [21]. The STEPS 2016 study also indicated a peak prevalence in the 65–69 age group [12]. Given the rapid aging of Iran's population [22], an increase in MetS among the elderly was expected. However, while some national surveys reported a decline in MetS among adults aged 25–65 years (from 35.95% in 2007 to 32.96% in 2011 based on IDF criteria) [23] the prevalence rose again to 43.5% in 2016 among adults aged ≥25 years [12]. Furthermore, a meta-analysis from 2000 to 2015 reported a non-significant decreasing trend in MetS prevalence among adults [24]. This trend, though not statistically significant, contrasts with our findings of a clear increase in MetS among older adults between 2016 and 2021. This discrepancy may be due to differences in the age groups studied, definitions used, or the broader temporal and demographic shifts in recent years. Our study specifically focuses on the elderly population, who face higher cardiometabolic risks and may have been underrepresented in previous analyses of the general adult population.

The prevalence of MetS in the elderly population is considerably high not only in Iran but also in other countries within the region and globally. For instance, using the NCEP ATPIII criteria, the prevalence of MetS among the elderly was reported as 49.5% in the United Arab Emirates (2015) [25], 71.5% in Qatar (2017) [26], 61.7% in Turkey (2011) [27], 59.0% in Brazil(2016) [28], and 63.02% in the United States(2018) [8]. A survey in Jordan revealed a high prevalence of MetS among the elderly using both IDF and NCEP ATPIII criteria, with an increasing trend from 2009 to 2017 [10]. Similarly, a growing trend has been observed in the United States, a developed country with a high-income economy. Data from the National Health and Nutrition Examination Survey (NHANES) showed a significant increase in MetS prevalence among the elderly, rising from 50.78% to 63.02% based on NCEP ATPIII criteria between 1999 and 2018 [8]. Therefore, the high prevalence of MetS among the elderly is not limited to a specific region. However, the observed differences may be influenced by the inconsistency of criteria used, particularly in the definition of central obesity based on the ethnic characteristics of each country.

Our study found that central obesity, HBP, and low HDL-C were the most prevalent components among the elderly MetS population, similar to findings from studies in Turkey [27] and Brazil [28]. Additionally, studies conducted in Jordan

**Table 3. National prevalence of MetS in Iranian elderly based on various definitions in 2016 and 2021 and its annual percent change.**

| | NCEP ATPIII | | | IDF | | | JIS | | |
|---|---|---|---|---|---|---|---|---|---|
| | **2016** | **2021** | **APC** | **2016** | **2021** | **APC** | **2016** | **2021** | **APC** |
| **Total** | 53.33 (51.03,55.61) | 60.12 (57.37,62.81) | 2.42 (1.19,3.71) | 59.8 (57.56,62) | 46.11 (43.81,48.44) | 54.15 (51.34,56.94) | 46.11 (43.81,48.44) | 54.15 (51.34,56.94) | 46.11 (43.81,48.44) |
| **Sex** | | | | | | | | | |
| Female | 68.75 (65.6,71.74) | 71.81 (68.12,75.23) | 0.89 (−0.48,2.25) | 76.18 (73.27,78.87) | 50.41 (47.08,53.73) | 57.37 (53.33,61.3) | 50.41 (47.08,53.73) | 57.37 (53.33,61.3) | 50.41 (47.08,53.73) |
| Male | 37.79 (34.72,40.96) | 48.22 (44.26,52.2) | 4.98 (2.56,7.5) | 43.3 (40.09,46.57) | 41.79 (38.58,45.07) | 50.88 (46.93,54.82) | 41.79 (38.58,45.07) | 50.88 (46.93,54.82) | 41.79 (38.58,45.07) |
| p-value | <0.001 | <0.001 | <0.001 | <0.001 | <0.001 | 0.023 | <0.001 | 0.023 | <0.001 |
| **Residence** | | | | | | | | | |
| Rural | 47.72 (44.56,50.9) | 54.06 (50.12,57.96) | 2.54 (0.48,4.55) | 53.32 (50.15,56.47) | 38.62 (35.56,41.78) | 46.47 (42.56,50.43) | 38.62 (35.56,41.78) | 46.47 (42.56,50.43) | 38.62 (35.56,41.78) |
| Urban | 56.13 (53.05,59.16) | 61.96 (58.58,65.23) | 1.99 (0.4,3.57) | 63.04 (60.08,65.9) | 49.86 (46.78,52.95) | 56.48 (53,59.88) | 49.86 (46.78,52.95) | 56.48 (53,59.88) | 49.86 (46.78,52.95) |
| p-value | <0.001 | 0.003 | 0.435 | <0.001 | <0.001 | <0.001 | <0.001 | <0.001 | <0.001 |

**NCEP ATPIII** National Cholesterol Education Program Adult Treatment Panel III; **IDF:** International Diabetes Federation; **JIS:** Joint Interim Statement; **AHA/NHLBI:** American Heart Association/National Heart, Lung, and Blood Institute; **RIDF:** Regional IDF; **RJIS:** regional JIS; **APC:** Annual Percent Change

All data are presented as percent (95% Confidence Interval)

[10] and the United States [8] identified central obesity and high FPG as the most rapidly increasing components. While our findings mostly align with these studies, we observed that the prevalence of central obesity remained unchanged, whereas elevated TG increased even more than high FPG. This discrepancy may arise because our study focused on elderly individuals with MetS, whereas the other studies examined changes in the general adult population. Nonetheless, our findings are consistent with the overall prevalence and trends of these metabolic risk factors in Iran as reported by other studies [29–32].

Low HDL-C is the most common lipid abnormality in Iran [30], as well as in other countries in the Middle East [33], while hypertriglyceridemia is the most common lipid abnormality in the United States [34]. In addition to genetic and ethnic predispositions, the high consumption of carbohydrate-rich foods, such as refined grains, in Iran may explain this variation [35]. Low HDL-C is the only lipid abnormality that declines constantly with age [30], although it remains among the highest in the elderly MetS population based on our study. Meanwhile, elevated TG is the most rapidly increasing component among the elderly MetS population. Hypertriglyceridemia has higher odds compared to other lipid abnormalities in the obese population [30], suggesting that this increase may be linked to the obesity epidemic. Furthermore, the prevalence of hypertriglyceridemia is higher in men [30], and our study showed that it was increasing at a faster rate in men, which may correlate with the greater increase in MetS in men compared to women.

One of the country's strategies to lower serum cholesterol has been the widespread prescription of statins by general practitioners [36], leading to a reduction in hypercholesterolemia [30] and this strategy should remain in practice. However, statins do not have the same effect on TG and HDL-C levels as they do on LDL-C and total cholesterol levels [37,38]. Therefore, interventions aimed at improving nutrition to reduce obesity by lowering trans-fat and sugar consumption, substituting carbohydrate-rich foods with protein-rich foods, increasing fruit and vegetable intake, and enhancing physical activity may be more beneficial for managing HDL-C and TG levels [39], as well as FPG levels, and consequently, MetS [40].

Elevated FPG is another component of MetS that our study found to be increasing. It is evident in literature that females are more susceptible to impaired glucose tolerance, obesity, and MetS [41]. In addition to that, global surveys revealed that while diabetes prevalence is generally similar between men and women, it is slightly higher in men under

| AHA/NHLBI | | | RIDF | | | RJIS | | |
|---|---|---|---|---|---|---|---|---|
| 2016 | 2021 | APC | 2016 | 2021 | APC | 2016 | 2021 | APC |
| 54.15 (51.34,56.94) | 67.76 (65.13,70.28) | 2.57 (1.51,3.67) | 46.11 (43.81,48.44) | 54.15 (51.34,56.94) | 3.26 (1.83,4.76) | 60.67 (58.43,62.87) | 69.79 (67.18,72.27) | 2.83 (1.8,3.91) |
| | | | | | | | | |
| 57.37 (53.33,61.3) | 79.15 (75.68,82.25) | 1.02 (−0.14,2.16) | 50.41 (47.08,53.73) | 57.37 (53.33,61.3) | 2.64 (0.65,4.63) | 68.58 (65.37,71.62) | 76.38 (72.79,79.63) | 2.19 (0.87,3.51) |
| 50.88 (46.93,54.82) | 56.16 (52.25,59.98) | 5.01 (2.94,7.19) | 41.79 (38.58,45.07) | 50.88 (46.93,54.82) | 4 (1.75,6.35) | 52.71 (49.53,55.86) | 63.07 (59.25,66.72) | 3.64 (1.92,5.44) |
| 0.023 | <0.001 | <0.001 | <0.001 | 0.023 | 0.13 | <0.001 | <0.001 | 0.096 |
| | | | | | | | | |
| 46.47 (42.56,50.43) | 59.72 (55.79,63.52) | 2.61 (0.75,4.42) | 38.62 (35.56,41.78) | 46.47 (42.56,50.43) | 3.79 (1.31,6.22) | 54.55 (51.39,57.67) | 62.13 (58.25,65.87) | 2.66 (0.89,4.38) |
| 56.48 (53,59.88) | 70.19 (66.97,73.22) | 2.09 (0.77,3.43) | 49.86 (46.78,52.95) | 56.48 (53,59.88) | 2.51 (0.71,4.33) | 63.74 (60.76,66.61) | 72.1 (68.89,75.09) | 2.49 (1.19,3.8) |
| <0.001 | <0.001 | 0.423 | <0.001 | <0.001 | 0.508 | <0.001 | <0.001 | 0.349 |

60. However, the prevalence rises more sharply in women after 65 [42], likely due to menopause [41,43]. These findings justify our observation of a higher prevalence of MetS and obesity in women, as well as a more pronounced upward trend in the high FPG component among elderly women.

From 2004 to 2021, BMI, the standardized mean WC, and the prevalence of obesity showed a statistically significant increasing trend in Iran. However, this trend was not significant for the prevalence of central obesity [31]. Correspondingly, our study found no significant increase in the central obesity component among the elderly MetS population.

Our findings revealed that among all components of MetS, the highest discrepancy in prevalence between females and males is observed in central obesity. Using international cut-points, as employed in the JIS (males: 94 cm, females: 80 cm), NCEP ATPIII, and AHA/NHLBI (males: 102 cm, females: 88 cm) criteria, females show a higher prevalence of central obesity. The highest prevalence was identified using JIS criteria, though the discrepancy between sexes was less pronounced compared to NCEP ATPIII and AHA/NHLBI criteria. Conversely, RJIS, which uses national cut-points (95 cm for both sexes) recommended by the National Committee of Obesity [20], shows a higher prevalence in males. This suggests that international cut-points may not accurately reflect the ethnic realities of Iran, leading to these paradoxical observations. Nonetheless, a higher prevalence of obesity according to BMI in females is evident [31].

In comparison to the global average and particularly developed countries, the disparity in obesity prevalence between sexes is more prominent in the Middle East and North Africa [44]. This region also has one of the highest parity rates worldwide, which correlates with an increased risk of obesity [45,46]. Additionally, the Middle East and North Africa have some of the worst physical activity levels globally, with a significantly higher prevalence of insufficient physical activity among women. Therefore, to address these issues, factors discouraging women from being physically active in the cultural context of this region should be mitigated, including personal challenges (lack of motivation, enjoyment, or skills in sports), absence of social support, environmental obstacles (limited free time or access to sports facilities), and cultural stigmas [44].

Our findings emphasize the growing need to recognize the elderly population as being at higher risk of NCDs due to the very high prevalence of MetS and its components. Policymakers in developing countries like ours, which are experiencing a demographic transition, need to consider the elderly as a growing group with unique needs. Integration of MetS

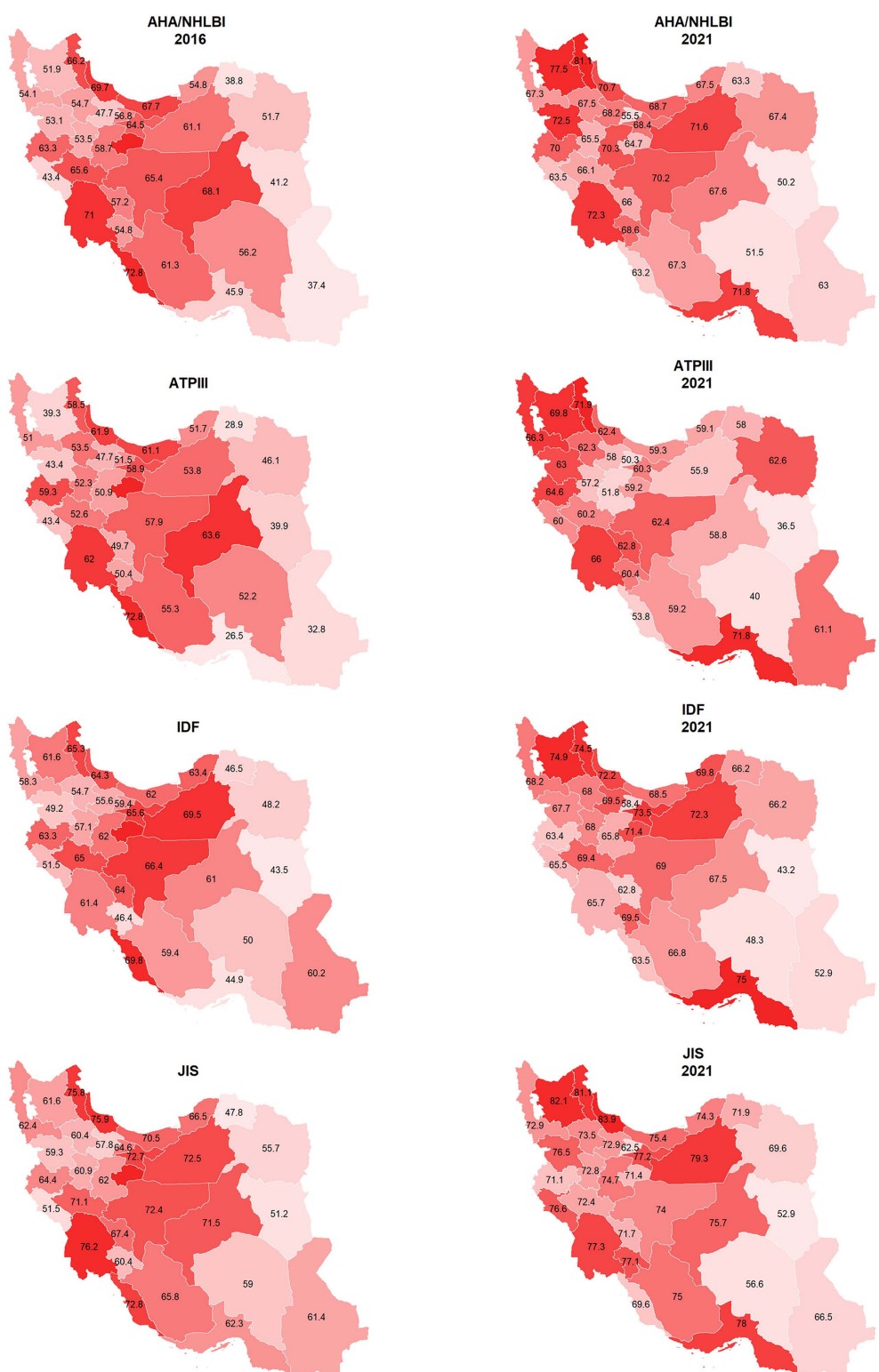

**Fig 1. Sub-national prevalence of MetS in Iranian elderly based on various MetS definitions in 2016 and 2021.** Administrative boundaries from GADM (Global Administrative Areas), www.gadm.org.

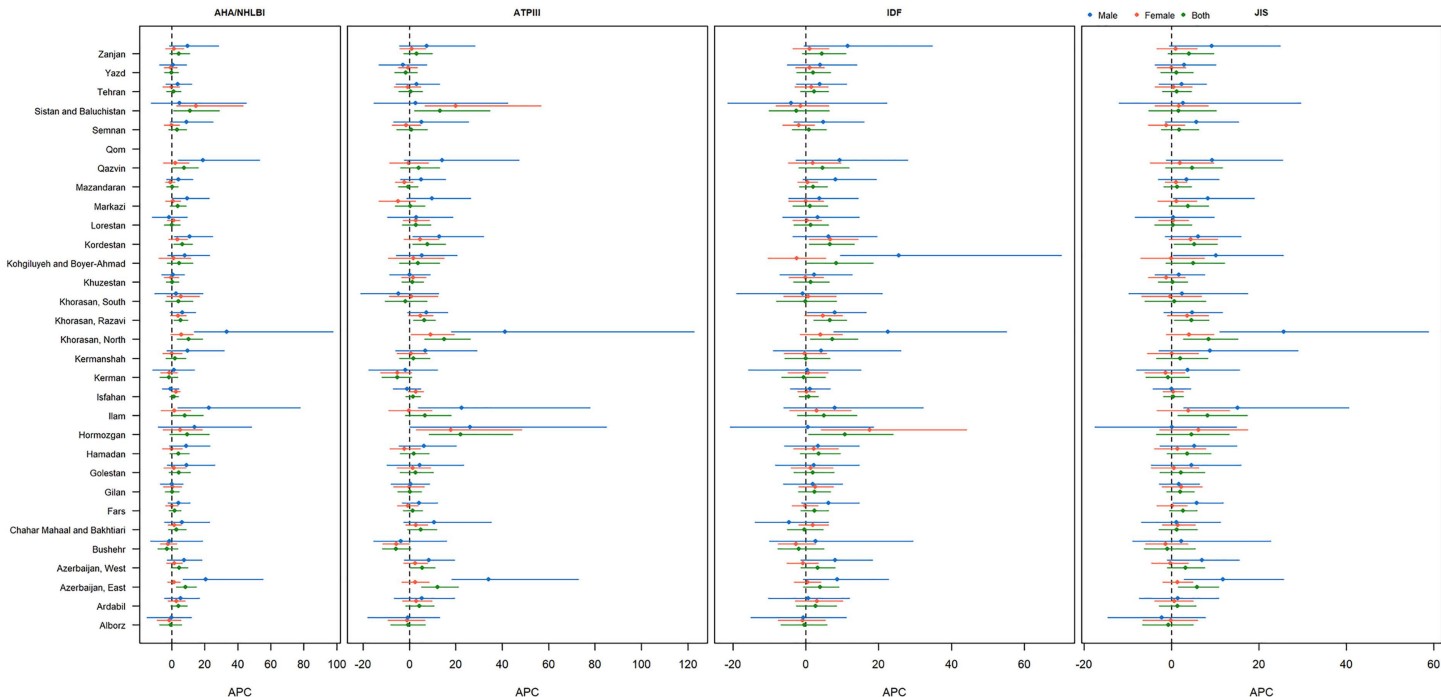

**Fig 2. Total and sex-specific sub-national annual percent change in prevalence of MetS in Iranian elderly from 2016 to 2021 based on various MetS definitions.** Administrative boundaries from GADM (Global Administrative Areas), www.gadm.org.

screening and management into routine geriatric care can facilitate early detection and treatment. Clinical guidelines tailored to older populations and equitable access to essential medications, especially for low-income groups, are also critical. Cross-sector collaboration involving health, social welfare, education, and urban planning is essential to address the multifactorial drivers of MetS. Furthermore, robust national monitoring and evaluation frameworks can inform evidence-based policymaking. However, policymakers must be cautious about labeling older people as a vulnerable group. Such labeling can be confused with frailty, dependence, and loss of autonomy, potentially fueling ageism and paternalistic benevolence towards the elderly, which is widely associated with poorer health outcomes [47]. Therefore, the common public perception that aging is an inevitable process of decline must be challenged. This can be achieved through education about healthy aging and promoting autonomy and self-determination among the elderly. Community-based prevention programs to empower and inform the elderly, along with provision of social network, promoting physical activity and dietary improvements, can be effective measures to reduce the burden of MetS and its components [48].

Our study has several strengths. First, we utilized large samples of elderly participants from nationally and sub-nationally representative STEPS surveys, enabling us to estimate the prevalence of MetS at the national level, as well as by sex and in both urban and rural areas across all provinces of Iran. Second, we portrayed the possible temporal changes in the prevalence of MetS by comparing results from the same study at two distinct time points. The data was collected using consistent methods and tools nationwide, with serum markers measured at a single center (NCDRC) under strict quality control. Third, we estimated the prevalence of MetS using all available definitions, including regionally modified ones, providing a framework for comparison with other studies regardless of the criteria used.

This study has several limitations. Firstly, the cross-sectional design of this study limits the ability to observe individual changes over time. Secondly, since the definition of some components of MetS in most criteria relied on both laboratory values and self-reported treatment status, the results may be subject to recall bias. Thirdly, the paucity of

**Table 4. National prevalence of MetS components in Iranian elderly with MetS based on various definitions in 2016 and 2021 and its annual percent change.**

| Component | | NCEP ATPIII | | | IDF | | | JIS | | |
|---|---|---|---|---|---|---|---|---|---|---|
| | | 2016 | 2021 | APC | 2016 | 2021 | APC | 2016 | 2021 | APC |
| Central Obesity | Total | 79.78 (77.48,81.9) | 79.45 (77.14,81.59) | −0.08 (−0.84,0.7) | – | – | – | 90.18 (88.56,91.6) | 90.7 (89.16,92.04) | 0.11 (−0.34,0.58) |
| | Female | 91.11 (89.06,92.82) | 91.87 (89.74,93.58) | 0.17 (−0.43,0.76) | – | – | – | 96.09 (94.72,97.12) | 96.64 (95.27,97.62) | 0.12 (−0.24,0.47) |
| | Male | 59.06 (54.41,63.56) | 59.39 (54.96,63.67) | 0.1 (−1.99,2.32) | – | – | – | 81.12 (77.68,84.13) | 82.49 (79.38,85.22) | 0.33 (−0.72,1.43) |
| | Urban | 80.7 (77.83,83.3) | 80.41 (77.71,82.86) | −0.08 (−1.02,0.87) | – | – | – | 90.05 (87.92,91.83) | 92.07 (90.36,93.51) | 0.44 (−0.12,1.01) |
| | Rural | 78.05 (74.1,81.55) | 76.99 (72.27,81.11) | −0.26 (−1.79,1.24) | – | – | – | 90.44 (87.77,92.57) | 87.21 (83.72,90.04) | −0.72 (−1.63,0.18) |
| Elevated FPG | Total | 67.28 (64.06,70.34) | 76.64 (73.59,79.43) | 2.64 (1.45,3.88) | 61.4 (58.29,64.42) | 69.71 (66.35,72.87) | 2.57 (1.21,3.99) | 62.4 (59.47,65.24) | 69.78 (66.61,72.78) | 2.26 (0.99,3.58) |
| | Female | 65.42 (61.28,69.34) | 75.71 (72.21,78.9) | 2.97 (1.42,4.6) | 59.32 (55.38,63.14) | 70.47 (66.54,74.11) | 3.51 (1.76,5.32) | 59.65 (55.79,63.39) | 70.3 (66.45,73.88) | 3.35 (1.64,5.11) |
| | Male | 70.68 (65.45,75.42) | 78.04 (72.17,82.97) | 1.99 (0.01,4.06) | 65.09 (60,69.86) | 68.57 (62.46,74.1) | 1.04 (−1.23,3.39) | 66.56 (62.08,70.76) | 69.1 (63.68,74.05) | 0.75 (−1.24,2.79) |
| | Urban | 68.15 (63.91,72.11) | 78.8 (75.15,82.03) | 2.94 (1.41,4.52) | 63.03 (58.95,66.93) | 71.06 (67.01,74.8) | 2.41 (0.7,4.17) | 64.15 (60.31,67.81) | 71.24 (67.38,74.82) | 2.11 (0.5,3.75) |
| | Rural | 65.22 (60.72,69.47) | 68.47 (63.36,73.17) | 1 (−1.04,2.97) | 57.55 (53.18,61.81) | 64.33 (59.32,69.04) | 2.27 (0.04,4.44) | 58.24 (54.1,62.27) | 64.25 (59.59,68.65) | 2.01 (−0.08,4.03) |
| Elevated TG | Total | 42.65 (39.47,45.89) | 57.67 (54.02,61.24) | 6.21 (4.22,8.32) | 51.14 (48.03,54.25) | 66.47 (63.22,69.57) | 5.38 (3.81,7.03) | 52.96 (50.02,55.87) | 67.68 (64.65,70.56) | 5.02 (3.6,6.52) |
| | Female | 41.13 (37.12,45.27) | 54.73 (49.96,59.41) | 5.89 (3.12,8.78) | 54.45 (50.64,58.2) | 67.79 (63.65,71.67) | 4.49 (2.59,6.46) | 55.18 (51.47,58.84) | 68.12 (64.08,71.91) | 4.31 (2.48,6.2) |
| | Male | 45.44 (40.26,50.72) | 62.12 (56.37,67.56) | 6.44 (3.42,9.72) | 45.28 (40.08,50.6) | 64.47 (59.15,69.45) | 7.3 (4.38,10.49) | 49.58 (44.88,54.3) | 67.09 (62.44,71.43) | 6.22 (3.82,8.8) |
| | Urban | 43.02 (38.94,47.19) | 57.45 (53.04,61.74) | 5.94 (3.37,8.63) | 52.16 (48.14,56.15) | 67.47 (63.64,71.08) | 5.27 (3.29,7.35) | 54.1 (50.31,57.83) | 68.45 (64.85,71.85) | 4.81 (3.01,6.68) |
| | Rural | 41.8 (37.27,46.47) | 58.49 (53.06,63.72) | 6.98 (3.91,10.07) | 48.74 (44.31,53.19) | 62.48 (57.18,67.51) | 5.12 (2.49,7.71) | 50.25 (46.04,54.46) | 64.73 (59.83,69.35) | 5.22 (2.84,7.56) |
| High Blood Pressure | Total | 85.84 (83.83,87.63) | 84.59 (82.52,86.45) | −0.29 (−0.91,0.35) | 89.2 (87.52,90.68) | 91.07 (89.5,92.42) | 0.41 (−0.05,0.89) | 89.59 (88.03,90.98) | 90.93 (89.43,92.23) | 0.29 (−0.15,0.75) |
| | Female | 85.16 (82.58,87.42) | 85.86 (83.26,88.12) | 0.17 (−0.64,0.98) | 89.25 (87.13,91.06) | 92.23 (90.27,93.82) | 0.66 (0.07,1.26) | 89.67 (87.62,91.41) | 92.29 (90.38,93.85) | 0.58 (0.01,1.16) |
| | Male | 87.07 (83.69,89.84) | 82.53 (78.93,85.63) | −1.07 (−2.13,0.02) | 89.11 (86.14,91.51) | 89.19 (86.38,91.47) | 0.01 (−0.81,0.86) | 89.48 (86.86,91.63) | 89.04 (86.47,91.17) | −0.1 (−0.84,0.66) |
| | Urban | 84.89 (82.3,87.15) | 84.16 (81.69,86.35) | −0.18 (−0.98,0.63) | 88.66 (86.52,90.5) | 90.55 (88.66,92.15) | 0.42 (−0.18,1.02) | 89.03 (87.02,90.75) | 90.38 (88.55,91.94) | 0.3 (−0.27,0.87) |
| | Rural | 87.59 (84.24,90.32) | 85.7 (81.58,89.02) | −0.43 (−1.56,0.69) | 90.19 (87.29,92.49) | 92.46 (89.39,94.69) | 0.51 (−0.33,1.32) | 90.64 (87.94,92.79) | 92.31 (89.44,94.45) | 0.37 (−0.41,1.13) |
| Reduced HDL-C | Total | 87.38 (85.1,89.35) | 84.21 (81.73,86.4) | −0.74 (−1.45,-0.01) | 87.39 (85.08,89.39) | 86.44 (83.8,88.71) | −0.22 (−0.95,0.53) | 88.23 (86.09,90.08) | 86.93 (84.48,89.04) | −0.3 (−0.97,0.39) |
| | Female | 87.44 (84.56,89.84) | 84.37 (81.28,87.02) | −0.7 (−1.62,0.2) | 89.35 (86.67,91.54) | 90.5 (88.14,92.43) | 0.26 (−0.47,1) | 89.82 (87.25,91.92) | 90.72 (88.43,92.6) | 0.2 (−0.5,0.91) |
| | Male | 87.27 (83.23,90.44) | 83.97 (79.59,87.55) | −0.77 (−2.01,0.5) | 83.92 (79.38,87.62) | 80.28 (74.75,84.85) | −0.88 (−2.46,0.72) | 85.84 (81.9,89.03) | 81.96 (77.19,85.92) | −0.92 (−2.25,0.43) |
| | Urban | 86.67 (83.68,89.18) | 83.62 (80.64,86.22) | −0.72 (−1.65,0.2) | 86.66 (83.58,89.24) | 86.65 (83.47,89.29) | −0.01 (−0.96,0.94) | 87.6 (84.74,89.98) | 87.1 (84.13,89.58) | −0.12 (−1,0.75) |
| | Rural | 89.04 (85.68,91.69) | 86.45 (82.21,89.8) | −0.58 (−1.7,0.53) | 89.12 (85.99,91.61) | 85.61 (81.2,89.12) | −0.79 (−1.94,0.32) | 89.74 (86.85,92.06) | 86.28 (82.27,89.51) | −0.77 (−1.82,0.24) |

NCEP ATPIII National Cholesterol Education Program Adult Treatment Panel III; **IDF:** International Diabetes Federation; **JIS:** Joint Interim Statement; **AHA/NHLBI:** American Heart Association/National Heart, Lung, and Blood Institute; **RIDF:** Regional IDF; **RJIS:** regional JIS; **APC:** Annual Percent Change

All data are presented as percent (95% Confidence Interval)

| AHA/NHLBI | | | RIDF | | | RJIS | | |
|---|---|---|---|---|---|---|---|---|
| 2016 | 2021 | APC | 2016 | 2021 | APC | 2016 | 2021 | APC |
| 76.16 (73.89,78.29) | 75.57 (73.28,77.71) | −0.16 (−0.96,0.66) | – | – | – | 75.47 (73.21,77.61) | 77.73 (75.56,79.76) | 0.59 (−0.19,1.39) |
| 89.06 (86.93,90.88) | 89.29 (87.06,91.17) | 0.06 (−0.59,0.7) | – | – | – | 73.43 (70.4,76.25) | 76.49 (73.53,79.21) | 0.83 (−0.28,1.94) |
| 53.64 (49.31,57.91) | 54.65 (50.55,58.68) | 0.36 (−1.79,2.63) | – | – | – | 78.22 (74.66,81.4) | 79.34 (76.02,82.3) | 0.28 (−0.88,1.49) |
| 76.85 (74.01,79.47) | 76.65 (73.99,79.12) | −0.06 (−1.05,0.93) | – | – | – | 78.18 (75.38,80.74) | 79.27 (76.74,81.59) | 0.27 (−0.66,1.21) |
| 74.86 (70.96,78.39) | 72.71 (68.1,76.88) | −0.56 (−2.16,1) | – | – | – | 70.45 (66.43,74.18) | 73.7 (69.28,77.68) | 0.92 (−0.72,2.51) |
| 64.46 (61.38,67.42) | 72.39 (69.36,75.23) | 2.34 (1.12,3.63) | 63.46 (59.94,66.85) | 71.52 (68.06,74.73) | 2.41 (1,3.9) | 65.38 (62.38,68.26) | 72.29 (69.34,75.07) | 2.03 (0.84,3.26) |
| 61.41 (57.48,65.21) | 71.84 (68.21,75.21) | 3.19 (1.56,4.89) | 62.55 (57.72,67.14) | 72.42 (68.19,76.29) | 2.98 (1.06,5) | 64.43 (60.37,68.3) | 73.52 (69.97,76.8) | 2.68 (1.11,4.33) |
| 69.72 (64.79,74.23) | 73.19 (67.74,78.01) | 0.97 (−0.97,2.98) | 64.57 (59.34,69.46) | 70.48 (64.77,75.6) | 1.76 (−0.43,4.06) | 66.63 (62.13,70.84) | 70.78 (65.78,75.32) | 1.21 (−0.66,3.15) |
| 65.26 (61.25,69.07) | 74.1 (70.47,77.42) | 2.56 (1,4.17) | 65.78 (61.23,70.05) | 72.45 (68.34,76.22) | 1.94 (0.18,3.75) | 66.93 (63.01,70.62) | 73.88 (70.34,77.13) | 1.99 (0.49,3.51) |
| 62.53 (58.17,66.7) | 65.78 (60.9,70.36) | 1.04 (−1.01,3.03) | 57.49 (52.33,62.49) | 67.74 (62.11,72.91) | 3.36 (0.87,5.82) | 61.77 (57.46,65.91) | 66.23 (61.44,70.71) | 1.43 (−0.61,3.4) |
| 57.8 (54.79,60.77) | 71.09 (68.02,73.97) | 4.22 (2.89,5.62) | 52.19 (48.65,55.71) | 66.99 (63.51,70.31) | 5.12 (3.42,6.92) | 56.04 (53.01,59.03) | 69.59 (66.57,72.45) | 4.42 (3.05,5.86) |
| 57.4 (53.65,61.06) | 69.91 (65.83,73.7) | 4.03 (2.26,5.86) | 58.17 (53.68,62.53) | 70 (65.43,74.2) | 3.78 (1.75,5.9) | 60.77 (56.93,64.48) | 71.9 (67.85,75.61) | 3.43 (1.73,5.17) |
| 58.51 (53.34,63.49) | 72.79 (68.04,77.06) | 4.45 (2.28,6.77) | 44.92 (39.58,50.38) | 63.54 (58.13,68.64) | 7.17 (4.15,10.46) | 49.85 (45.12,54.58) | 66.74 (62.12,71.06) | 6 (3.6,8.58) |
| 58.98 (55.12,62.74) | 71.3 (67.65,74.7) | 3.86 (2.17,5.6) | 53.67 (49.2,58.09) | 67.67 (63.58,71.51) | 4.73 (2.62,6.95) | 57.32 (53.42,61.14) | 70.06 (66.47,73.42) | 4.08 (2.35,5.88) |
| 54.96 (50.47,59.38) | 70.27 (65.24,74.85) | 5.06 (2.81,7.28) | 48.36 (43.15,53.6) | 64.27 (58.26,69.87) | 5.88 (2.89,8.89) | 53.05 (48.62,57.44) | 67.8 (62.8,72.42) | 5.05 (2.73,7.35) |
| 90.55 (88.94,91.94) | 91.63 (90.1,92.94) | 0.24 (−0.21,0.69) | 88.92 (86.96,90.61) | 91.15 (89.42,92.62) | 0.5 (−0.03,1.04) | 90.68 (89.1,92.05) | 91.3 (89.78,92.61) | 0.13 (−0.3,0.59) |
| 90.23 (88.15,91.98) | 92.76 (90.84,94.3) | 0.56 (−0.02,1.13) | 88.83 (86.11,91.07) | 92.92 (90.69,94.65) | 0.91 (0.2,1.64) | 91.51 (89.44,93.2) | 93.09 (91.18,94.62) | 0.35 (−0.21,0.91) |
| 91.1 (88.38,93.23) | 89.9 (87.18,92.1) | −0.27 (−1.02,0.51) | 89.03 (85.98,91.49) | 88.93 (86.03,91.3) | −0.03 (−0.87,0.85) | 89.56 (86.94,91.71) | 88.97 (86.37,91.12) | −0.14 (−0.88,0.63) |
| 90.33 (88.31,92.03) | 91.16 (89.31,92.72) | 0.18 (−0.38,0.75) | 88.96 (86.57,90.97) | 90.54 (88.45,92.29) | 0.35 (−0.31,1.01) | 90.39 (88.39,92.07) | 90.79 (88.94,92.35) | 0.08 (−0.48,0.65) |
| 90.96 (88.1,93.19) | 92.85 (89.83,95.03) | 0.42 (−0.4,1.21) | 88.82 (85.12,91.69) | 92.86 (89.4,95.25) | 0.9 (−0.09,1.87) | 91.21 (88.42,93.38) | 92.64 (89.68,94.79) | 0.32 (−0.48,1.09) |
| 90.98 (89.03,92.61) | 90.14 (88.16,91.83) | −0.19 (−0.73,0.38) | 87.4 (84.88,89.55) | 86.73 (84.2,88.91) | −0.16 (−0.89,0.61) | 89.8 (87.78,91.52) | 88.45 (86.39,90.23) | −0.3 (−0.89,0.3) |
| 90.75 (88.19,92.8) | 91.98 (89.78,93.73) | 0.27 (−0.4,0.95) | 90.38 (87.78,92.48) | 90.9 (88.08,93.1) | 0.12 (−0.65,0.89) | 92.75 (90.81,94.31) | 92.34 (90.11,94.1) | −0.08 (−0.67,0.49) |
| 91.37 (88.08,93.81) | 87.51 (83.67,90.55) | −0.86 (−1.85,0.15) | 83.78 (79.11,87.57) | 81.95 (77.47,85.71) | −0.45 (−1.85,1.01) | 85.93 (81.97,89.13) | 83.65 (79.88,86.82) | −0.54 (−1.69,0.65) |
| 90.75 (88.2,92.8) | 89.89 (87.51,91.86) | −0.2 (−0.9,0.51) | 87.65 (84.42,90.29) | 86.66 (83.66,89.19) | −0.23 (−1.17,0.7) | 89.64 (86.97,91.81) | 88.36 (85.91,90.44) | −0.29 (−1.04,0.46) |
| 91.52 (88.56,93.77) | 91.12 (87.61,93.71) | −0.08 (−0.98,0.8) | 86.77 (82.71,89.99) | 87.01 (82.35,90.59) | 0.07 (−1.23,1.34) | 90.17 (87.14,92.54) | 88.77 (85.01,91.68) | −0.3 (−1.28,0.66) |

STEPS studies that include the elderly population restricts our ability to accurately comment on the temporal trends in MetS prevalence. Although our study includes some statistical analysis to evaluate changes, further investigations using additional national surveys are needed to more clearly define the trends in MetS prevalence within this age group. This underscores the need for dedicated studies to better understand and address the specific health challenges of the elderly population.

## Conclusions

MetS is a significant health challenge among the elderly, characterized by an alarmingly high prevalence and a likely increasing trend. Although our findings are based on Iranian data, they may be relevant to other low- and middle-income countries with similar epidemiological and demographic transitions, ethnic predispositions, and limited economic capacity. Consequently, prioritizing community gerontology programs aimed at promoting healthy aging and empowering the elderly population is crucial. Central obesity, HBP, and low HDL-C are the most prevalent components of MetS, with notable upward trends observed in elevated TG and high FPG. This underscores the need for targeted interventions to promote healthy nutrition, lifestyles, and physical activity among the elderly. Given the higher prevalence of MetS and obesity in females, efforts should focus on removing cultural barriers and discrimination that hinder females' physical activity. Future research should continue to explore temporal trends and comparative analyses across similar demographic groups to inform effective public health strategies.

## Supporting information

**S1 Table. Sub-national prevalence of MetS in Iranian elderly based on various MetS definitions in 2016 and 2021 and its annual percent change.**
(PDF)

**S1 File. STROBE-checklist.**
(DOCX)

## Author contributions

**Data curation:** Seyed Mohammad-Navid Ataei, Ali Golestani, Sepehr Khosravi, Mohammad-Mahdi Rashidi, Ozra Tabatabaei-Malazy, Rosa Haghshenas, Bagher Larijani.

**Formal analysis:** Ali Sheidaei.

**Methodology:** Ali Sheidaei, Kazem Khalagi.

**Supervision:** Ozra Tabatabaei-Malazy.

**Validation:** Ali Sheidaei, Mohammad-Mahdi Rashidi, Ozra Tabatabaei-Malazy, Rosa Haghshenas.

**Writing – original draft:** Seyed Mohammad-Navid Ataei.

**Writing – review & editing:** Seyed Mohammad-Navid Ataei, Ali Sheidaei, Ali Golestani, Sepehr Khosravi, Mohammad-Mahdi Rashidi, Ozra Tabatabaei-Malazy, Rosa Haghshenas, Kazem Khalagi, Bagher Larijani.

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
