## [Decision Letter · Decision Letter 0]

16 Jul 2025

Dear Dr. Tabatabaei-Malazy,

Thank you for submitting your manuscript to PLOS ONE. After careful consideration, we feel that it has merit but does not fully meet PLOS ONE’s publication criteria as it currently stands. Therefore, we invite you to submit a revised version of the manuscript that addresses the points raised during the review process.

This is a well-written article. Some points have been raised by the reviewers, please modify the article accordingly and resubmit the article. It is also recommended to edit the article for language as there are several grammatical mistakes in the text

We look forward to receiving your revised manuscript.

Kind regards,

Patricia Khashayar

Academic Editor

PLOS ONE

Journal Requirements:

2. In the online submission form, you indicated that the data are available upon request.

4. Please remove all personal information, ensure that the data shared are in accordance with participant consent, and re-upload a fully anonymized data set.

Reviewers' comments:

Reviewer's Responses to Questions

**Comments to the Author**

1. Is the manuscript technically sound, and do the data support the conclusions?

Reviewer #1: Yes

Reviewer #2: Yes

2. Has the statistical analysis been performed appropriately and rigorously?

Reviewer #1: Yes

Reviewer #2: Yes

3. Have the authors made all data underlying the findings in their manuscript fully available?

Reviewer #1: No

Reviewer #2: Yes

4. Is the manuscript presented in an intelligible fashion and written in standard English?

Reviewer #1: No

Reviewer #2: Yes

Reviewer #1: I reviewed the manuscript entitled “Over-time changes in the prevalence of metabolic syndrome and its components among elderly population in Iran from 2016 to 2021; a nation-wide study”. It is an interesting topic. The manuscript is in the scope of the journal, but certain shortcomings should be addressed before the article can be published:

• The manuscript should be edited for language, as it contains several grammatical mistakes.

• In the abstract, Add a brief info on inclusion and exclusion.

• In the Introduction, it is highly suggested to add the prevalence of Mets in Iran as well as worldwide.

• In the Method section:

o Why did authors not use Iranian criteria for diagnostic for MetS such as Esteghamati or Azizi studies?

• The results section needs more explanation.

• At the beginning of the discussion, the most important results of the study should be written and better to merge the first two paragraphs together. The reason behind the difference noticed between this study and other studies should be explained in more detail.

• I strongly advise you to provide more strengths as well as the limitations of the study in the discussion part.

• The second table is a bit confusing; it is recommended to separate the frequency and mean subsequently.

• As it is requesting to publish in an international multidisciplinary Journal, in the conclusion you should include how the findings of this study can affect the general practice except for Iran ethnic groups population and some insight on future studies. The generalizability of the results should also be discussed.

• It is highly recommended that citations be added to the figures.

Reviewer #2: This is a well-conducted study that uses nationally representative STEPS survey data to provide robust evidence on temporal trends and component patterns of metabolic syndrome among the elderly population in Iran. Here are some comments to further improve the clarity and depth of the manuscript.

1. Abstract:

Please include the total number of study participants in the abstract, along with brief demographic details of your study population.

2. Introduction:

While the introduction provides background on the global burden of metabolic syndrome and its components, it does not clearly articulate the specific research gap this study aims to address. Please consider adding a sentence or two explaining what is currently unknown about MetS trends in the Iranian elderly population or what previous studies have "not" covered.

3. Discussion:

Lines 317-325, the cited studies refer to different age ranges and population scopes. These differences limit the comparability of findings. The paragraph cites a meta-analysis showing a decreasing (though statistically non-significant) trend in MetS prevalence, yet concludes that the trend is accelerating. This contradiction should be discussed with more nuance.

The policymaking section highlights key strategies but needs more depth. Consider adding recommendations on community-based prevention, clinical guidelines, equitable medication access, integration into geriatric care, cross-sector collaboration, and national monitoring frameworks.

**Do you want your identity to be public for this peer review?** For information about this choice, including consent withdrawal, please see our Privacy Policy

Reviewer #1: **Yes: ** Pouria Khashayar

Reviewer #2: No

---

## [Author Response · Author response to Decision Letter 1]

12 Aug 2025

Dear Editor and Reviewers,

We would like to sincerely thank you and the reviewers for your valuable comments and suggestions which have greatly improved the quality of our manuscript. We have carefully revised the manuscript in accordance with the comments provided. Below, we provide a detailed point-by-point response to each of the reviewers’ and editor’s comments.

Reviewer 1:

1. The manuscript should be edited for language, as it contains several grammatical mistakes.

We appreciate the reviewer’s feedback. The manuscript has undergone professional English editing to address all grammatical issues and improve clarity and readability.

2. In the abstract, Add a brief info on inclusion and exclusion.

Thank you for the suggestion. We have revised the abstract to include the inclusion and exclusion criteria. We have also added a sentence in the Study Design subsection of the Methods to briefly describe the exclusion criteria used in the STEPs survey.

3. In the Introduction, it is highly suggested to add the prevalence of Mets in Iran as well as worldwide.

We appreciate the reviewer’s suggestion. The Introduction already included national estimates of MetS prevalence in Iran based on the 2016 national survey, indicating a prevalence range of 32% to 47.6%, with higher rates among females, urban residents, and the elderly. To further clarify the context and emphasize the relevance of our study, we have added a sentence highlighting the lack of recent nationwide data specifically focused on the elderly population and the limited evidence on temporal trends in MetS prevalence in Iran.

4. In the Method section:

Why did authors not use Iranian criteria for diagnostic for MetS such as Esteghamati or Azizi studies?

Thank you for raising this important point. As described in the Methods section, we applied the Iranian National Committee of Obesity’s recommendation to use a uniform waist circumference (WC) cut-off of 95 cm for both men and women. This locally adapted threshold was integrated into the IDF and JIS definitions, resulting in two regionally modified criteria: Regional IDF (RIDF) and Regional JIS (RJIS). These approaches, which are based on the work of Esteghamati and Azizi et al. , allowed us to reflect population-specific characteristics while preserving international comparability. We have cited the relevant reference accordingly.

5. The results section needs more explanation.

We appreciate the reviewer’s suggestion. While we did not make substantial changes to the Results section, we would like to clarify our approach. The main findings were already organized under clear subsections (Participant Characteristics, Overall Prevalence of MetS, Sex and Residence Differences, Sub-national Disparities, and Components of MetS) to facilitate readability and highlight key outcomes. Given the inclusion of detailed data in the accompanying tables, we deliberately avoided extensive textual repetition to maintain focus and avoid unnecessary length or complexity in the narrative.

Should the reviewer have specific areas in mind that require further elaboration, we would be happy to address them in a future revision.

6. At the beginning of the discussion, the most important results of the study should be written and better to merge the first two paragraphs together. The reason behind the difference noticed between this study and other studies should be explained in more detail.

Thank you for this valuable comment. We have revised the beginning of the Discussion section to emphasize the key findings of our study in the first paragraph. To improve clarity and coherence, the second and third paragraphs have been merged. We now begin the discussion with a summary of previous studies showing that the prevalence of MetS increases with age, providing context and justification for the high prevalence observed in our elderly population. We then discuss prior research on temporal trends in MetS prevalence and compare these trends with our own findings. The differences between our results and those of previous studies are now addressed in greater detail, with emphasis on the fact that our study specifically focused on the elderly population, which may explain some of the observed discrepancies.

7. I strongly advise you to provide more strengths as well as the limitations of the study in the discussion part.

Thank you for your recommendation. In the current version of the manuscript, we have already included two comprehensive paragraphs in the Discussion section that detail the key strengths and limitations of our study. These cover aspects such as the use of large nationally representative samples, standardized data collection and laboratory analysis, and the application of multiple diagnostic definitions, as well as limitations related to the cross-sectional design, reliance on self-reported data, and limited availability of elderly-specific STEPS surveys.

We believe these sections adequately reflect the methodological rigor and constraints of the study. However, if the reviewer has any specific point in mind that requires further elaboration, we would be happy to address it in the next revision.

8. The second table is a bit confusing; it is recommended to separate the frequency and mean subsequently.

Thank you for the suggestion. Table 2 has been reformatted to clearly distinguish frequency distributions from continuous variables with means and standard deviations.

9. As it is requesting to publish in an international multidisciplinary Journal, in the conclusion you should include how the findings of this study can affect the general practice except for Iran ethnic groups population and some insight on future studies. The generalizability of the results should also be discussed.

We agree with the reviewer. The Conclusion section has been revised to discuss the broader implications of our findings and their relevance for international audiences and future studies.

10. It is highly recommended that citations be added to the figures.

Thank you for your suggestion. All figures presented in the manuscript are generated from our own analysis of the national STEPS survey data. As such, they are original and not derived from external sources.

Reviewer 2:

1. Please include the total number of study participants in the abstract, along with brief demographic details of your study population.

Thank you for your recommendation. We have revised the abstract to include the total sample size and key demographic characteristics of the study population.

2. While the introduction provides background on the global burden of metabolic syndrome and its components, it does not clearly articulate the specific research gap this study aims to address. Please consider adding a sentence or two explaining what is currently unknown about MetS trends in the Iranian elderly population or what previous studies have "not" covered.

Thank you for this valuable comment. A clear statement of the research gap has been added at the end of the Introduction to emphasize what this study adds to the existing literature.

3. Lines 317-325, the cited studies refer to different age ranges and population scopes. These differences limit the comparability of findings. The paragraph cites a meta-analysis showing a decreasing (though statistically non-significant) trend in MetS prevalence, yet concludes that the trend is accelerating. This contradiction should be discussed with more nuance.

We appreciate the reviewer’s observation. This paragraph has been revised to clearly state the differences in populations and methods across studies and to address the inconsistency in trend interpretations.

4. The policymaking section highlights key strategies but needs more depth. Consider adding recommendations on community-based prevention, clinical guidelines, equitable medication access, integration into geriatric care, cross-sector collaboration, and national monitoring frameworks.

We agree and have expanded the section to include additional recommendations for policy and practice, including community-based prevention strategies, access to medication, and integration into geriatric care.

Editor’s Technical Requirements:

We have followed the PLOS ONE guidelines for formatting and have appropriately named all uploaded files.

2. In the online submission form, you indicated that the data are available upon request.

All PLOS journals now require all data underlying the findings described in their manuscript to be freely available to other researchers, either 1. In a public repository, 2. Within the manuscript itself, or 3. Uploaded as supplementary information. This policy applies to all data except where public deposition would breach compliance with the protocol approved by your research ethics board. If your data cannot be made publicly available for ethical or legal reasons (e.g., public availability would compromise patient privacy), please explain your reasons on resubmission and your exemption request will be escalated for approval.

The datasets generated and/or analyzed during the current study are not publicly available due to the restrictions set by the funder of the main STEPS project, I.R. Iran’s National Institute of Health Research, but are available from the corresponding author on reasonable request.

We appreciate the suggestion. All figures in the manuscript present original analyses and visualizations produced by the study team as part of this study.

4. Please remove all personal information, ensure that the data shared are in accordance with participant consent, and re-upload a fully anonymized data set.

The presented tables/figures are without any personal information. As we said above, the datasets generated and/or analyzed during the current study are not publicly available due to the restrictions set by the funder of the main STEPS project, I.R. Iran’s National Institute of Health Research, but are available from the corresponding author on reasonable request.

Captions for all Supporting Information files have been added to the end of the manuscript, and in-text citations have been updated accordingly.

The reviewers did not recommend any specific references to be cited, and no changes were made to the reference list in this regard.

The reference list has been reviewed for completeness and accuracy. No retracted articles were cited.

Sincerely,

Authors

---

## [Editor Report · Decision Letter 1]

19 Aug 2025

Over-time changes in the prevalence of metabolic syndrome and its components among elderly population in Iran from 2016 to 2021; a nation-wide study

PONE-D-25-11109R1

Dear Dr. Tabatabaei-Malazy,

We’re pleased to inform you that your manuscript has been judged scientifically suitable for publication and will be formally accepted for publication once it meets all outstanding technical requirements.

Kind regards,

Patricia Khashayar

Academic Editor

PLOS ONE
---

## [Editor Report · Acceptance letter]

PONE-D-25-11109R1

PLOS ONE

Dear Dr. Tabatabaei-Malazy,

I'm pleased to inform you that your manuscript has been deemed suitable for publication in PLOS ONE. Congratulations! Your manuscript is now being handed over to our production team.

Kind regards,

on behalf of

Dr. Patricia Khashayar

Academic Editor

PLOS ONE